# Assessment of Dyspnoea, Physical Activity, and Back Pain Levels in Students at Medical Universities after the COVID-19 Pandemic in Poland

**DOI:** 10.3390/jpm13101474

**Published:** 2023-10-08

**Authors:** Monika Gałczyk, Anna Zalewska, Marek Sobolewski

**Affiliations:** 1Faculty of Health Sciences, University of Lomza, 14 Akademicka St., 18-400 Lomza, Poland; aanna.zalewska@gmail.com; 2Plant of Quantitative Methods, Rzeszow University of Technology, Al, Powstancow Warszawy 12, 35-959 Rzeszow, Poland; mareksobol@poczta.onet.pl

**Keywords:** post COVID-19, dyspnoea, physical activity, back pain, students

## Abstract

Objectives: The purpose of this research was to assess the extent of dyspnoea, physical activity (PA), and back pain complaints and the association of dyspnoea, PA, and back pain complaints with PA in post-COVID-19 students at medical universities in Poland. Methods: An online survey was carried out among Polish medical students (213 women and 204 men) who had had a positive test for SARS-CoV-2 within the last year. The Medical Research Council (MRC) dyspnoea scale was used to assess the degree of dyspnoea. The International Physical Activity Questionnaire (IPAQ) was used to determine the level of PA. The Oswestry Disability Index (ODI) and the Neck Disability Index (NDI) were used to assess back discomfort. Results: The study group had average levels of PA, with median total activity significantly lower in women (median total activity for women was 1189 and for men was 2044, while the standard deviation for women was 1419 and for men was 1450). More than 93% of the students reported no symptoms of dyspnoea. The following results were observed for ODI (median of 1.2 for women and 1.7 for men and standard deviation of 3.1 for women and 4.0 for men) and for NDI (median of 2.8 for women and 2.5 for men, standard deviation of 4.3 for women and 4.0 for men). Cervical spine pain was more frequent and severe. There are small, statistically significant correlations between the MRC and IPAQ measures and the ODI and NDI and IPAQ. Conclusions: In the study group of students of medicine, dyspnoea linked with a history of COVID-19 is not an issue. Post-pandemic PA levels should be increased in this group, with particular attention to female students. Urgent measures are also needed to prevent cervical pain in students at medical universities in Poland.

## 1. Introduction

Although the COVID-19 pandemic no longer poses a risk to the general public in everyday life, there is growing evidence of coronavirus complications in those who have overcome the disease but have not fully recovered. A considerable proportion of this population is made up of young people with no prior health history [1,2,3].

There is already substantial evidence in the literature on the adverse health effects of SARS-CoV-2 on patients. However, researchers and scientists are becoming increasingly interested in the issue of subacute and long-term post-COVID-19 symptoms, which can affect a variety of systems and organs, such as the autonomic system through chest pain, tachycardia, and palpitations; the respiratory system through dyspnoea, cough, and general fatigue; and the musculoskeletal system through muscle and joint pain [4,5,6,7,8,9].

As a result, a general term has emerged in the literature for patients who have recovered from COVID-19 but still have some post-viral symptoms that last longer than four weeks and are unaccounted for by other diagnoses. The National Institute for Health and Care Excellence (NICE), the Scottish Intercollegiate Guidelines Network, and the Royal University of General Practitioners issued guidelines. Fatigue, chronic dyspnoea, muscle weakness, and decreased levels of physical activity (PA) are the most typically reported symptoms. In the present situation, it is necessary to spread knowledge about the long-term consequences of COVID-19 among both medical personnel and the general population [10,11,12,13,14,15,16,17].

Dyspnoea is a subjective impression of having trouble breathing, although it is a common and uncomfortable symptom experienced by patients, according to the American Thoracic Society. In individuals suffering from a variety of diseases, it can occur with little exertion or even at rest. In healthy or young people, on the other hand, it usually occurs with heavy exertion but is treated as a normal physical response [18,19,20].

Most studies available in the literature show that PA levels declined sharply both during and after the COVID-19 pandemic, not only in children but also in adults. One of the main reasons for this was the change in lifestyle from active to passive, spending more time in front of the computer and smartphone, which often led to a change in eating habits and a reduction in the static–dynamic load-bearing capacity of the spine, which in turn can contribute to spinal pain [21,22,23,24,25,26,27,28,29]. 

The COVID-19 pandemic’s long-term consequences can reverberate at both the physical and psychological levels. One of the professional groups most involved in the fight against the coronavirus was healthcare workers, who were often assisted in their work by medical students. In the context of caring for critically ill and often dying patients, they were at a particularly high risk of becoming infected with SARS-CoV-2 [30,31,32].

Therefore, the authors of this study decided to survey medical students in Poland and perform a preliminary assessment of dyspnoea severity, physical activity PA, and prevalence of back pain after the COVID-19 pandemic. The authors also asked two additional research questions: "Is there a link between the research group’s level of dyspnoea and PA?” and “Is there an association between back pain complaints and PA among medical university trainees in post-pandemic Poland COVID-19?”

## 2. Material and Methods

### 2.1. Participants and Procedure

An online cross-sectional survey was carried out in June 2023 among Polish students at medical universities who had been diagnosed with COVID-19 at least one year ago. Self-report measures of dyspnoea, PA, and back pain were obtained using an online Google Forms questionnaire. In groups of medical university students in Poland, a request to take part in the study with a link to the questionnaire, information about the study, and agreement to participate was posted on social media. The data collection form was designed in such a way that it was not possible to proceed to the next question without answering one of the previous questions. The inclusion criteria for this study were as follows: medical student status (medicine, dentistry, nursing, midwifery, physiotherapy, emergency medicine, pharmacy, medical analytics, electroradiology, dietetics, speech therapy, cosmetology, public health or dental technology), at least one year since diagnosis of COVID-19, and age above 18 years. The exclusion criteria were the presence of chronic respiratory, cardiac, or musculoskeletal diseases. The authors received responses from 417 students aged 19–25 years who met the inclusion criteria. The study group was gender homogeneous.

This study was carried out in accordance with the Helsinki Declaration principles and was approved by the Senate Committee on Ethics in Scientific Research of the University of Medical Sciences in Bialystok (KB/18/2020.2021).

### 2.2. Methods of Assessing the Level of Dyspnoea, Physical Activity, and Back Pain

**Table 1 jpm-13-01474-t001:** Methods used to measure the studied features.

Examined Feature	Tool
Dyspnoea	Medical Research Council Dyspnoea Scale (MRC)
Physical Activity	International Physical Activity Questionnaire (IPAQ)
Pain in the cervical spine	Neck Disability Index (NDI)
Pain in the lumbar spine	Oswestry Disability Index (ODI)

#### 2.2.1. Medical Research Council Dyspnoea Scale (MRC)

The MRC dyspnoea scale [33] was used to evaluate the level of dyspnoea (Table 1). It comprises 5 questions, with 0 indicating dyspnoea on exertion only and 4 indicating dyspnoea at rest. The authors of this study used the scale in accordance with the National Council of Physiotherapists guidelines for the treatment and application of rehabilitation of patients after COVID-19 [34]. The Cronbach’s alpha value for the MRC is >0.7 [35].

#### 2.2.2. International Physical Activity Questionnaire (IPAQ)

To assess the level of PA, the authors used the Polish-language short version of the IPAQ (Table 1), which is appropriate for adults aged 15 to 69 years. It contains seven questions describing all levels of PA in daily life and is expressed in the units of MET (metabolic equivalent of work) /min/week [36]. The Cronbach’s alpha value for the IPAQ is >0.7 [37,38,39].

#### 2.2.3. Oswestry Disability Index (ODI), Neck Disability Index (NDI)

The spinal level of discomfort was evaluated by the authors using the ODI questionnaire, which assesses the lumbar spine as well as the NDI questionnaire, which assesses the cervical spine (Table 1). Both tools are concerned with assessing the degree of disability experienced by people with spinal pain in everyday life. They each contain 10 questions, and the answers are scored as follows: A—0 points, B—1 point, C—2 points, D—3 points, E—4 points, and F—5 points. The maximum number of points a patient can score is 50. The disability rating scale is as follows:None—0–4;Small—5–14 points;Average—15–24 points;Serious—25–34 points;Total—over 35 points.

The Cronbach’s alpha value for ODI and NDI is >0.7 [40,41].

### 2.3. Statistical Methods

The distribution of PA measures at a given level of intensity, total effort, and complaints of back pain were presented using selected descriptive statistics. The comparison of the PA level of subjects with and without dyspnoea consisted of comparing the positional statistics of the activity measures in both groups and assessing the difference between them using the Mann–Whitney test separately for women and men. The separation by gender was important because the measurements of physical activity are completely different for men and women; otherwise, the results of the analyses would have been completely different. The non-parametric Mann–Whitney test was chosen since the distributions of the IPAQ, ODI, and NDI measures were non-normal (high skewness and kurtosis, with *p* < 0.001 for the Shapiro–Wilk test). Spearman’s rank correlation coefficient was used to examine the correlation between PA and pain intensity. It was dictated by the strong asymmetry in the distribution of the measures compared (particularly, ODI and NDI) and the fact that the relationship between activity and pain complaints is not necessarily linear.

For statistical analysis, Statistica v. 13 software (TIBCO Software Inc., Palo Alto, CA, USA (2017)) was applied. A significance level of *p* < 0.05 (*) was established for all statistical analyses; however, the results for *p* < 0.01 (**) and *p* < 0.001 (***) were additionally denoted for all statistical analyses. 

## 3. Results

### 3.1. Characteristics of the Sample Population

The analysis involved a cohort of 417 medical university students who had been diagnosed with COVID-19 for at least one year, with the highest response rates in medicine (19%), physical therapy (16%), emergency medicine (13%), and dentistry (12%). The study population consisted of almost equal numbers of women and men (48.9% and 51.1%, respectively) aged 19 to 25 years (Table 2). Due to the homogeneity in the study group in terms of maturity, this feature was not taken into account in subsequent analyses, although gender specificity was taken into account when defining the distribution of psychometric measures and their connection.

### 3.2. Assessment of Physical Activity Levels

PA levels were assessed using the IPAQ questionnaire. The distribution of measures of PA with a given intensity level and total effort is presented using selected descriptive statistics (Table 3). The table includes a breakdown by gender, as the PA of women and men is at a different level (median total activity for women is 1189 and for men is 2044). 

The IPAQ scores were also classified into a three-degree adjectival scale, which is shown in the table below (Table 4). 

### 3.3. Assessing the Level of Dyspnoea

Dyspnoea was assessed using the five-point MRC scale. Dyspnoea scores of 0, 1, and 2 were observed in the study population. The two highest scores of 3 and 4 were not observed.

The distribution of dyspnoea scores, broken down by student gender, is shown in Table 5. As can be seen, the extent of dyspnoea is almost identical in males and females, and furthermore, no symptoms were observed in the vast majority of respondents (more than 93% of students reported no symptoms of dyspnoea). Those who reported any type of discomfort, with one exception, reported a minimum level of dyspnoea severity-1.

Due to such a rare occurrence of dyspnoea, only the dichotomous division between those without and with dyspnoea symptoms will be used in further analyses.

### 3.4. Pain Complaints

The vast majority of students reported no pain in the lumbar spine (338 to be exact, or 81.1% had an ODI of 0 points) and almost half reported no pain in the cervical spine area (202 to be exact, or 48.4% of students had an NDI of 0 points) (Table 6). Among men, 79.4% had no low back pain and 47.1% had no cervical pain. Among women, these percentages were slightly higher: 82.6 and 49.8%, respectively. With such severe disparity in the distribution of ODI and NDI readings, the comparison between men and women cannot be based on positional statistics, which are usually 0 points, but must be based on the mean. There was hardly any difference in pain levels between men and women. It can also be noticed that pain in the cervical spine is more frequent and more severe.

### 3.5. Physical Activity Level and Somatic Complaints

#### 3.5.1. Activity and Occurrence of Dyspnoea

A comparison was made between the levels of PA for people with and without dyspnoea (Table 7). The results are somewhat surprising, as it turns out that those who reported the presence of some symptoms of dyspnoea are those who are more physically active. At the same time, this relationship holds for all activity intensities for both men and women.

#### 3.5.2. Activity and the Incidence of Pain

Correlations between the PA of the students surveyed and the intensity of pain complaints were examined (Table 8). Lumbar back pain was more common in those who were more physically active—this correlation occurred between both groups of men and women. The correlations between the ODI and all IPAQ measures were statistically significant, although their strength was rather low (coefficients just above 0.30)—the greater the PA, the greater the complaints of low back pain. Significantly, no such correlations were found between the NDI and the IPAQ. In the women’s group, no correlation was found between the PA and the NDI value, while in the men’s group, a weak negative correlation was marked—less physically active men suffer from cervical spine pain.

## 4. Discussion

The health benefits associated with maintaining adequate levels of PA (from cardiovascular to musculoskeletal to mental health) are well known [42]. In contrast, the consequences of a lack of PA are among the primary risk factors for non-communicable diseases (NCDs) worldwide [43] and influence the more difficult course of infectious diseases [43]. Therefore, maintaining proper levels of PA is a critical element in maintaining health, especially after COVID-19 and during prolonged COVID-19 therapy [42,43,44]. During the pandemic, a drop in overall PA was noticed among university students worldwide due to epidemic restrictions [45,46,47].

In the study population of medical students, the majority present an average level of PA, while the median total activity is significantly lower for women (median total activity of 1189 for women and 2044 for men). For example, a 2016 study found that more than half of medical students in the US adhered to pre-pandemic guidelines for prescribed levels of PA [48], which changed significantly during the pandemic [45,46,47]. Gender differences in activity levels have long been known and should be given special attention [49,50,51]. During the pandemic, it was the men in the medical student group who had higher MET *w*/*w* scores compared with the women, which could also be related to the different frequency of intense PA among men. During the pandemic, most women did not have intensive PA, which affects the overall activity level [52]. Overall, 22.1% of women surveyed by the authors of this study had low levels of PA, which is consistent with studies conducted during the pandemic that showed an increase in the number of physically inactive medical students [52]. According to the literature, this level has not yet returned to its pre-pandemic levels, when about half of the medical students were physically active as reported by [48].

In the studies presented here, the level of dyspnoea is almost identical in men and women, and furthermore, no symptoms of dyspnoea were observed in most respondents (over 93% of the students). Those who reported any kind of discomfort indicated that it was dyspnoea, with the exception of one case. This is largely consistent with the findings of a study in which the frequency and severity of dyspnoea after COVID-19 were correlated with age, time since onset of SARS-CoV-2 infection symptoms, and the presence of comorbidities, among other factors [19].

Unfortunately, an increase in the frequency of low back complaints in young people has been noted in the literature, which is directly related to the COVID-19 pandemic and associated limitations, such as distance learning [53].

In our study, the majority of students had no lumbar back pain (81.1 per cent had an ODI of 0 points) and almost half had no cervical complaints (48.4 per cent of students had an NDI of 0 points). There was little difference between the sexes in the extent of lumbar pain, but cervical pain was found to be more frequent and severe. Our observations are in complete agreement with other results of a study among medical students conducted following the COVID-19 pandemic [54]. This trend (especially in terms of higher severity of cervical pain) was observed in medical students during the pandemic by Gomes et al. [55], as well as by other authors among students from different fields of study worldwide [56,57]. It seems that this is related, for example, to the prolonged daily use of electronic products, an inappropriate sitting posture, a persistent tilting of the head and the increasing observation of the so-called text neck phenomenon in young people [58,59]. Remaining in a static position for a long time while studying is one of the crucial risk factors for musculoskeletal complaints. The simultaneous adoption of an abnormal posture, as observed by Gomes et al. in a study of medical students, exacerbates this phenomenon [55]. Part of the literature also mentions female gender as a risk factor for cervical pain in young people, which contradicts the results of our own study [56,58].

In addition, we examined the levels of PA in patients with and without dyspnoea. The results are somewhat surprising as it seems that those who reported having some symptoms of dyspnoea were more physically active. At the same time, this relationship holds for all intensities of PA in both men and women. There seems to be an inversion of the cause–effect relationship, i.e., a small group of people who report dyspnoea are aware that it may be a consequence of a lack of PA and undertake it more often. In addition, physically active people may experience discomfort from dyspnoea during PA (especially at high intensity), while those with low activity may not have the opportunity to notice any discomfort in themselves [60].

Lumbar pain was found to be more common in people with higher levels of PA—this correlation was found in both men and women. The correlations between the ODI and all IPAQ measures were statistically significant, although their strength was rather low (coefficients of slightly above 0.30). That is, the more active the students surveyed were, the more they complained of lower back pain. According to the authors of this study, the explanation for this seemingly surprising correlation could lie in two facts—first, physically inactive people do not have pain complaints due to inactivity at a young age, but this is not a health advantage because in a person who leads an active life, a certain amount of pain complaints is a normal physiological reaction of the body [61]. Especially in the lower back, this discomfort may simply be the result of physical exertion [62,63]. The second fact is that the IPAQ and the ODI and NDI are determined based on subjective responses of the respondents and that physically active people may place more emphasis on accuracy in completing the ODI and NDI. It is significant that such correlations with the IPAQ do not exist for the NDI, as cervical pain is more associated with time spent lying down and sitting, especially in front of a computer screen [58]. In the women’s group, no correlation was found between PA and the NDI score, while in the men’s group, there was a weak negative correlation—less physically active men have slightly more cervical pain.

The study presented here has limitations that the authors acknowledge. The first is the cross-sectional nature of this study on a small group of students (who, by their nature, are not able to provide robust and causal evidence for the observed relationships) The second limitation is the online survey. The third is that the questionnaires are designed to provide results based on the respondents’ self-assessments. The survey also has strengths, the most important of which is the use of standardised survey instruments. The survey was also conducted in a short time and did not require a large financial cost. Further surveys should be conducted in a larger population, using more objective approaches to analyse the items collected.

## 5. Conclusions

Dyspnoea associated with a history of COVID-19 is not a problem in the study group of medical university students. The level of post-pandemic PA should be increased in this group, especially in female students. Urgent measures are also needed to prevent cervical pain in medical university students in Poland.

## Figures and Tables

**Table 2 jpm-13-01474-t002:** Age and gender of students surveyed.

Gender	Number	Percentage
man	204	48.9%
woman	213	51.1%
Age (Years)	Number	Percentage
19	47	11.3%
20	123	29.5%
21	98	23.5%
22	80	19.2%
23	49	11.8%
24	17	4.1%
25	3	0.7%

**Table 3 jpm-13-01474-t003:** Level of PA as measured using the IPAQ questionnaire.

IPAQ Measures	Mean	Std. Dev.	Median	Low Quartile	Upper Quartile	Min	Max	Skewness	Kurtosis
Males									
Intense effort	907	712	840	320	1280	0	3840	0.90	0.78
Moderate effort	577	446	480	160	800	0	2400	1.17	1.68
Walking	599	506	528	132	908	0	2772	1.27	2.48
Total effort	2082	1419	2044	678	3068	132	7616	0.75	0.36
Females									
Intense effort	615	480	615	0	960	0	2800	1.06	0.93
Moderate effort	428	360	373	160	640	0	2000	1.19	1.89
Walking	590	495	495	198	825	0	2772	1.25	2.36
Total effort	1633	1450	1189	612	2420	0	5355	0.86	0.49

IPAQ—International Physical Activity Questionnaire.

**Table 4 jpm-13-01474-t004:** Classification of IPAQ scores.

Activity Level	Gender	
	Male	Woman
low	20 (9.8%)	47 (22.1%)
average	123 (60.3%)	139 (65.3%)
high	61 (29.9%)	27 (12.7%)

**Table 5 jpm-13-01474-t005:** Distribution of dyspnoea scores.

MRC	Gender		Total
	Man	Woman	
0	190 (93.1%)	200 (93.9%)	390 (93.5%)
1	14 (6.9%)	12 (5.6%)	26 (6.2%)
2	0 (0.0%)	1 (0.5%)	1 (0.2%)

MRC—Medical Research Council Dyspnoea Scale.

**Table 6 jpm-13-01474-t006:** Spinal pain complaints.

Back Pain	Mean	Std. Dev.	Median	Low Quartile	Upper Quartile	Min	Max	Skewness	Kurtosis
Males									
ODI	1.7	4.0	0	0	0	0	27	2.87	9.74
NDI	2.5	3.7	1	0	3	0	16	1.64	1.65
Females									
ODI	1.2	3.1	0	0	0	0	17	2.99	9.34
NDI	2.8	4.3	1	0	4	0	20	1.77	2.53

ODI—Oswestry Disability Index; NDI—Neck Disability Index.

**Table 7 jpm-13-01474-t007:** Level of PA and prevalence of dyspnoea among respondents.

IPAQ Measures	Occurrence of Dyspnoea						*p*
	No			Yes			
	*N*	Median	IQR	*N*	Me	IQR	
Males							
Intense effort	190	720	960	14	1440	960	0.0026 **
Moderate effort	190	480	640	14	760	480	0.0252 *
Walking	190	495	693	14	792	594	0.0083 **
Total effort	190	1935	2321	14	3208	1556	0.0037 **
Females							
Intense effort	200	440	960	13	1280	640	0.0000 ***
Moderate effort	200	360	440	13	640	200	0.0022 **
Walking	200	495	627	13	825	363	0.0007 ***
Total effort	200	1374	1751	13	2712	1088	0.0000 ***

IQR—inter-quartile range; the *p*-value was calculated using Mann–Whitney test; *p* < 0.05 (*); *p* < 0.01 (**); *p* < 0.001 (***); IPAQ—International Physical Activity Questionnaire.

**Table 8 jpm-13-01474-t008:** Results of correlation analysis between measures of PA and intensity of back pain complaints.

IPAQ	Pain Complaints			
	Males		Females	
	ODI	NDI	ODI	NDI
Intense effort	0.36 (*p* = 0.0000 ***)	−0.11 (*p* = 0.1063)	0.36 (*p* = 0.0000 ***)	0.02 (*p* = 0.8139)
Moderate effort	0.36 (*p* = 0.0000 ***)	−0.15 (*p* = 0.0360 *)	0.30 (*p* = 0.0000 ***)	−0.02 (*p* = 0.7703)
Walking	0.31 (*p* = 0.0000 ***)	−0.19 (*p* = 0.0070 **)	0.22 (*p* = 0.0011 **)	−0.12 (*p* = 0.0819)
Total effort	0.37 (*p* = 0.0000 ***)	−0.18 (*p* = 0.0093 **)	0.33 (*p* = 0.0000 ***)	−0.08 (*p* = 0.2335)

The table shows Spearman’s rank correlation coefficient values together with an assessment of the statistical significance of the analysed relationships; *p* < 0.05 (*); *p* < 0.01 (**); *p* < 0.001 (***); IPAQ—International Physical Activity Questionnaire; ODI—Oswestry Disability Index; NDI—Neck Disability Index.

## Data Availability

The data presented in this study are available on request from the authors.

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
