# Peer review of "Assessment of Dyspnoea, Physical Activity, and Back Pain Levels in Students at Medical Universities after the COVID-19 Pandemic in Poland"

_jpm, 2023, doi:10.3390/jpm13101474_

Round 1

Reviewer 1 Report

Dear Researchers, 

please kindly consider the following points: 

1- The title could be more specific about the population of medical students that was studied. For example, the title could specify the gender, age, and year of study of the medical students who were included in the study. 

2- The abstract could be more specific about the methods that were used to collect data. For example, the abstract could specify the questionnaires or surveys that were used to collect data on dyspnea, physical activity, and back pain. The abstract could be more specific about the results of the study. For example, the abstract could specify the mean and standard deviation of the MRC Dyspnoea Scale scores, IPAQ scores, ODI scores, and NDI scores.

3- The Introduction should be expanded. its too summarized. review of literature should be more comepletly stated. the following are recommended: 

Taheri, Morteza, et al. "Effects of home confinement on physical activity, nutrition, and sleep quality during the COVID-19 outbreak in amateur and elite athletes." Frontiers in nutrition 10 (2023): 1143340.

Sharif, Mohammad Reza, and Mansour Sayyah. "Assessing Physical and Demographic Conditions of Freshman." International journal of Sport Studies for Health 1.1 (2018).

4-  Methods section could include a table that summarizes the different measures that were used to assess dyspnea, physical activity, and back pain. The methods section could include a flow diagram that summarizes the steps that were involved in the data collection and analysis process. The methods section could include a discussion of the limitations of the study design and the data collection methods.

5-  The use of figures would make the results section more visually appealing and easier to understand. It would also help to make the results more memorable for readers.

6- The discussion could be more specific about the implications of the findings for future research. For example, the authors could discuss how their findings could be used to develop interventions to improve physical activity levels in medical students or to reduce the incidence of dyspnea and back pain in medical students. The discussion could be more specific about the limitations of the study. For example, the authors could discuss how the cross-sectional design of the study limits the ability to make causal inferences about the relationship between physical activity and dyspnea and back pain.

Author Response

Dear Reviewer,

Thank you for your very valuable comments on the article. We are very grateful to you for taking the time to assess our manuscripts and for their constructive comments.
Please find attached the revised version of the manuscript and a „Point-by-point response to reviewers” file, explaining the revisions made.
We kindly inform you that the introduced changes to the manuscript have been highlighted indicated by using Tracked changes.
We hope that edited version meets the standards of the Journal and we are looking forward to hearing from you.
Yours sincerely,
Monika Gałczyk
Anna Zalewska
Marek Sobolewski

# Reviewer 1
1- The title could be more specific about the population of medical students that was studied. For example, the title could specify the gender, age, and year of study of the medical students who were included in the study.
The authors are grateful for the reviewer's suggestions. Due to the fact that students of various medical faculties (not only medicine) were included in the study, the authors decided to use a different general term - students of medical universities. The group is homogeneous in terms of gender, and the age range of the respondents includes students of all years of study.
2- The abstract could be more specific about the methods that were used to collect data. For example, the abstract could specify the questionnaires or surveys that were used to collect data on dyspnea, physical activity, and back pain. The abstract could be more specific about the results of the study. For example, the abstract could specify the mean and standard deviation of the MRC Dyspnoea Scale scores, IPAQ scores, ODI scores, and NDI scores.
The authors only listed the questionnaires used in the abstract so as not to exceed the number of words permitted by law. Mean and standard deviation were provided for the IPAQ, ODI and NDI measures.
3- The Introduction should be expanded. its too summarized. review of literature should be more comepletly stated. the following are recommended:
Taheri, Morteza, et al. "Effects of home confinement on physical activity, nutrition, and sleep quality during the COVID-19 outbreak in amateur and elite athletes." Frontiers in nutrition 10 (2023): 1143340.
Sharif, Mohammad Reza, and Mansour Sayyah. "Assessing Physical and Demographic Conditions of Freshman." International journal of Sport Studies for Health 1.1 (2018).
The authors are grateful for the remarck. The introduction has been corrected and suggested literature has been added.
4- Methods section could include a table that summarizes the different measures that were used to assess dyspnea, physical activity, and back pain. The methods section could include a flow diagram that summarizes the steps that were involved in the data collection and analysis process. The methods section could include a discussion of the limitations of the study design and the data collection methods.
A table summarizing the various measurements used to assess shortness of breath, physical activity, and back pain has been added to the methods section. The authors included limitations related to the designed study and data collection methods at the end of the discussion, therefore they no longer included the same elements in the methods section to prevent duplication of information. They also did not decide on a flow chart due to the large number of tables in the manuscript and the reluctance to repeat information found in the text.
5- The use of figures would make the results section more visually appealing and easier to understand. It would also help to make the results more memorable for readers.
The authors thank the reviewer for his remarck. The authors have made every effort to ensure that the tables presented in the text are legible to the reader. When writing subsequent articles, they will take this into account and present the obtained data in a different graphic form.
6- The discussion could be more specific about the implications of the findings for future research. For example, the authors could discuss how their findings could be used to develop interventions to improve physical activity levels in medical students or to reduce the incidence of dyspnea and back pain in medical students. The discussion could be more specific about the limitations of the study. For example, the authors could discuss how the cross-sectional design of the study limits the ability to make causal inferences about the relationship between physical activity and dyspnea and back pain.
The authors thank the reviewer for his remarck. The discussion has been improved based on reviewer suggestions.

Reviewer 2 Report

The paper presented the assessment of the extent of dyspnoea, physical activity and back pain complaints, and the association of dyspnoea and physical activity and back pain complaints with physical activity in post-COVID-19 medical students in Poland.

 Regarding the study's strengths, it is important to note that interesting sample of Polish medical students (213 women and 204 men) who had a positive test for SARS-CoV-2 within the last year has been chosen.

The topic of the manuscript is within the scope of the Journal and could be relatively valuable to the scientific audience.

 The quality of the research design is acceptable. The title of the article is accurate.

Abstract reflects the work done and the conclusions drawn.

Some clarifications are however needed:

INTRODUCTION

The background of empirical studies (2023), which justify the present study, should be updated; and this same thing has to be compared in discussion and conclusions.

 METHOD

The section is missing some important information.

Representativeness – statistical power (sample size calculation), should be presented.

Please provide explanation why nonparametric statistics methods have been chosen.

Please explain why the Mann-Whitney test separately for women and men has been used?

 RESULTS

Why are the values of kurtosis not shown in Table 2 and Table 3? Not only skewness but also kurtosis should be reported. 

DISCUSSION AND CONCLUSIONS

 Results should be explained based on the answer to the research question, and in comparison, with other very significant and current studies (2023).

Please discuss the nonparametric effect size of your study. Were the observed effects strong/mild/weak? Compare the effect sizes in your data with the effect sizes in previous studies if possible.

TO SUM UP I think the author(s) need to make the recommended corrections.

Author Response

Dear Reviewer,

Thank you for your very valuable comments on the article. We are very grateful to you for taking the time to assess our manuscripts and for their constructive comments.
Please find attached the revised version of the manuscript and a „Point-by-point response to reviewers” file, explaining the revisions made.
We kindly inform you that the introduced changes to the manuscript have been highlighted indicated by using Tracked changes.
We hope that edited version meets the standards of the Journal and we are looking forward to hearing from you.
Yours sincerely,
Monika Gałczyk
Anna Zalewska
Marek Sobolewski

# Reviewer 2 The paper presented the assessment of the extent of dyspnoea, physical activity and back pain complaints, and the association of dyspnoea and physical activity and back pain complaints with physical activity in post-COVID-19 medical students in Poland. Regarding the study's strengths, it is important to note that interesting sample of Polish medical students (213 women and 204 men) who had a positive test for SARS-CoV-2 within the last year has been chosen. The topic of the manuscript is within the scope of the Journal and could be relatively valuable to the scientific audience. The quality of the research design is acceptable. The title of the article is accurate. Abstract reflects the work done and the conclusions drawn. The authors thank you for appreciating the strengths of the manuscript. Some clarifications are however needed:

INTRODUCTION The background of empirical studies (2023), which justify the present study, should be updated; and this same thing has to be compared in discussion and conclusions.

The authors thank the reviewer's remarck and updated the background of empirical studies.

METHOD The section is missing some important information.

Representativeness – statistical power (sample size calculation), should be presented.

As several psychometric measures (ODI, NDI, IPAQ) were used in the study, it was difficult to establish clear criteria for the accuracy of the results and to determine the sample size on this basis. For this reason, it was decided to focus on the accuracy of the estimate of dyspnoea prevalence. Assuming a confidence level of 90%, an expected prevalence of dyspnoea of 10% and a confidence interval of +/-3%, the required sample size was to be 370 people. We were able to obtain data from 417 people, so we can assume that the reported prevalence of dyspnoea was estimated with an accuracy of at least +/-3%.

Please provide explanation why nonparametric statistics methods have been chosen.

The need to use non-parametric methods, in particular the Mann-Whitney test for comparing the two groups or the Spearman rank correlation for analysing the relationship, was due to the very strong asymmetry in the distribution of the measures of activity (IPAQ) and pain complaints ODI and NDI. In the article, a rationale was given for the choice of the Spearman rank correlation coefficient ("due to the strong asymmetry in the distribution of the compared measures (especially ODI and NDI) and the fact that the relationship between activity and pain complaints is not necessarily linear"), and in the current version an analogous rationale was added for the choice of the Mann-Whitney test.

Please explain why the Mann-Whitney test separately for women and men has been used?

Distinction by gender breakdown was necessary because the physical activity measures for women and men are at completely different levels - analysing the data for women and men together could also lead to an overestimation of the variability in the physical activity measures, which would reduce the possibility of detecting significant differences due to the presence of dyspnoea (in Table 6).

RESULTS

Why are the values of kurtosis not shown in Table 2 and Table 3?

Not only skewness but also kurtosis should be reported. Thank you for this remark - the kurtosis values are given in Tables 2 and 5. They confirm the very large deviations from the normal distribution for the numerical characteristics analysed, clearly indicating the need to choose non-parametric statistical methods.

DISCUSSION AND CONCLUSIONS

Results should be explained based on the answer to the research question, and in comparison, with other very significant and current studies (2023). Please discuss the nonparametric effect size of your study. Were the observed effects strong/mild/weak? Compare the effect sizes in your data with the effect sizes in previous studies if possible.

The discussion was improved by recent literature and the information that the correlations between ODI, NDI and physical activity were very weak. However, the authors would like to add that no such comparisons have been made in the literature, so there is no reference point - our work will provide such a reference point

TO SUM UP I think the author(s) need to make the recommended corrections.

Thank you, the authors tried to make corrections

Round 2

Reviewer 1 Report

ACCEPTED